# Fabrication and Characterization of a Flexible Non-Enzymatic Electrochemical Glucose Sensor Using a Cu Nanoparticle/Laser-Induced Graphene Fiber/Porous Laser-Induced Graphene Network Electrode

**DOI:** 10.3390/s25072341

**Published:** 2025-04-07

**Authors:** Taeheon Kim, James Jungho Pak

**Affiliations:** School of Electrical Engineering, Korea University, Seoul 136-713, Republic of Korea; page21c@korea.ac.kr

**Keywords:** laser-induced graphene, flexible sensor, Cu nanoparticle, non-enzymatic electrochemical detection

## Abstract

We demonstrate a flexible electrochemical biosensor for non-enzymatic glucose detection under different bending conditions. The novel flexible glucose sensor consists of a Cu nanoparticle (NP)/laser-induced graphene fiber (LIGF)/porous laser-induced graphene (LIG) network structure on a polyimide film. The bare LIGF/LIG electrode fabricated using an 8.9 W laser power shows a measured sheet resistance and thickness of 6.8 Ω/□ and ~420 μm, respectively. In addition, a conventional Cu NP electroplating method is used to fabricate a Cu/LIGF/LIG electrode-based glucose sensor that shows excellent glucose detection characteristics, including a sensitivity of 1438.8 µA/mM∙cm^2^, a limit of detection (LOD) of 124 nM, and a broad linear range at an applied potential of +600 mV. Significantly, the Cu/LIGF/LIG electrode-based glucose sensor exhibits a relatively high sensitivity, low LOD, good linear detection range, and long-term stability at bending angles of 0°, 45°, 90°, 135°, and 180°.

## 1. Introduction

Flexible electronic devices have recently attracted considerable attention because of their potential application as wearable devices for direct monitoring of physiological signals in ubiquitous healthcare systems [1,2,3,4,5,6]. Among various flexible biosensors, electrochemical sensors have been widely used for detecting vital signs, mainly because they have many advantages such as a simple fabrication method, stable signals, rapid response, and broad applicability [7,8,9,10,11,12]. Due to the significant demand for the self-diagnosis of human metabolic conditions, many flexible sensors have been applied in glucose detection. As is well known, glucose is a key indicator of human metabolic condition [13,14]. In addition, glucose monitoring can provide noninvasive and easy access to metabolic information in human sweat, for example, enabling the diagnosis of diabetes mellitus, hypoglycemia, and other diseases [15,16,17,18]. Thus, flexible glucose sensors can be conformally attached to the curved skin surface of individuals for real-time health condition monitoring with accurate, stable, and fast responses.

Electrochemical glucose biosensors may be divided into enzyme-based and non-enzyme-based sensors. The enzyme-based glucose sensors generally utilize glucose oxidase (GO_x_) or glucose dehydrogenase (GDH), which are commercially used due to their higher selectivity and faster assay compared to optical and Raman spectroscopy sensing techniques [19]. However, the enzyme-based glucose sensors have various inherent drawbacks associated with enzymes, such as electrode surface fouling, enzyme denaturation, weak enzyme immobilization on the electrode, or complex enzyme purification processes [20,21]. These drawbacks can degrade the sensitivity, reliability, or accuracy of the sensors. In contrast, non-enzymatic glucose sensors are capable of directly oxidizing glucose on the electrode surface without the need for enzymes. This feature favors non-enzymatic glucose sensors for glucose analysis because of their affordability, enhanced sensitivity, excellent selectivity, and stability.

To realize non-enzymatic glucose sensing applications, copper is widely used as a catalyst to induce oxidation reactions with glucose in an alkaline electrolyte; it also shows outstanding catalytic functionality on micro- and nanostructured electrodes with modified carbon-based materials such as carbon nanotubes (CNTs), graphene, and reduced graphene oxide [22,23,24,25,26]. In addition, micro- and nanostructured Cu nanoparticles (NPs) can be easily electroplated electrochemically and can have tremendous advantages such as being low cost, biocompatible, environmentally friendly, having enhanced mass transport, a high surface area, and an improved signal-to-noise ratio (S/N) [26,27,28]. In this study, Cu NPs were electrochemically deposited on the surface of a laser-induced graphene fiber (LIGF) and porous laser-induced graphene (LIG) network for glucose detection.

Porous LIG and LIGF, developed by the Tour group, can be fabricated by irradiating a commercial polyimide (PI) film surface with an infrared CO_2_ laser. Specifically, porous LIG patterned in vector mode (the *x*–*y* direction) is formed at locally high temperatures (>2500 °C), which decrease the nitrogen and oxygen content by breaking the C–O, C=O, and N–C bonds [29]. LIGF patterned in raster mode (the *x* direction) can be grown by overlapping the laser spots so that the overlapped laser spots can lead to overheating and destruction of the upper LIG layers, resulting in the remaining LIG layers to have a thickness of <50 μm [30]. This LIGF employed in sensor electrodes provides a large active surface area, resulting in improved sensor sensitivity.

LIG as a sensor electrode offers outstanding advantages, including low cost, design flexibility, durability, reproducibility, one-step fabrication, and physical flexibility. Moreover, due to its porous structure, excellent conductivity, and electrochemical stability, the LIG-based electrode has recently been adopted in electrochemical sensing applications for lead ion [31], phytotoxic aluminum ion [32], nitrite [33], phosphate [34], dopamine [35], bisphenol A [36], SARS-CoV-2 [37], H_2_O_2_ [38], NO_2_ [39], glucose [40,41,42], and so on. As demonstrated by these many studies, LIG can be applied in various sensing platforms with its chemical modification and functionalization.

In terms of the commercial aspects of the disposable blood glucose sensor market, LIG electrodes have significant advantages. Currently, the dominant SPCEs (screen-printed carbon electrodes) in the glucose sensor market are low cost and suitable for mass production. However, according to several published papers, the screen-printing process requires stencils, conductive solutions, and heat treatment, incurring costs each time the electrode design is changed [43,44]. In contrast, the production of LIG electrodes requires only PI film as a raw material, allows for easy design changes, and has the potential to be easily applied in a roll-to-roll process [45].

In this paper, we used a method for fabricating sensor electrodes similar to that reported by the Tour group [29,30] and optimized the electrical properties of the LIGF/LIG electrode by using different laser powers in the raster laser patterning mode. The LIGF/LIG electrode was fabricated by a direct laser process from a polyimide (PI) substrate. The electrodes consist of 3 electrodes (working, counter, and reference) for an electrochemical sensor. The LIGF/LIG working electrode was then functionalized by Cu NP electroplating for glucose detection. The LIGF/LIG counter and reference electrodes are bare, while the LIGF/LIG reference electrode is simply coated with Ag/AgCl ink. Our fabricated sensor exhibits excellent characteristics, including sensitivity, limit of detection (LOD), selectivity, linearity, and reliability over a range of glucose concentrations. In addition, we also investigated the electrochemical properties of the Cu NP/LIGF/LIG-based glucose sensor under bending angles of 0°, 45°, 90°, 135°, and 180°.

## 2. Experimental Section

### 2.1. Materials and Apparatus

A 125-μm-thick Kapton (PI) film was purchased from Isoflex (Korea) and used as the precursor material for the LIGF electrode. K_3_[FE(CN)_6_], KNO_3_, CuSO_4_, Na_2_SO_4_, NaOH, D-glucose, ascorbic acid (AA), uric acid (UA), lactose, fructose, and sucrose were purchased from Sigma-Aldrich Co. (St. Louis, MO, USA). All the other reagents were of analytical grade and used without further purification. Solutions were made using Millipore water (≈18 MΩ).

A digital multimeter (Agilent U1272A, Santa Clara, CA, USA) was used for measurement and calculation of the resistance and sheet resistance of the fabricated LIGF/LIG electrodes. A potentiostat (PC4/750, Gamry Instruments, Warminster, PA, USA) was used to measure all the electrochemical reactions of the fabricated glucose sensor. In the three-electrode system of our fabricated glucose sensor, the working electrode (WE) and the counter electrode (CE) were composed of LIGF/LIG with Ag/AgCl ink (BAS Inc., Tokyo, Japan) as the reference electrode (RE).

Scanning electron microscopy (SEM) was performed using a Quanta 250 FEG instrument manufactured by FEI (Thermo Fisher Scientific, Waltham, MA, USA). X-ray diffraction (XRD) patterns were obtained using a SmartLab instrument (Cu Kα1, *λ* = 1.54 Å, ICDD card #00-041-1487, Rigaku, Tokyo, Japan). A LabRAM ARAMIS IR^2^ (HORIBA JOBIN YVON, Kyoto, Japan) Raman microscope system was used under 532 nm laser excitation and a 5 mW laser power at room temperature (RT). X-ray photoelectron spectroscopy (XPS) was performed using an X-TOOL instrument (ULVAC-PHI, Chigasaki, Japan) under a 15 kV aluminum source power, 100 µm beam size, and 140 eV scan energy.

### 2.2. Preparation of LIGF/LIG Electrode

The three-electrode system (WE, CE, and RE) of the glucose sensor was designed using AutoCAD 2018 (Autodesk Inc., San Francisco, CA, USA). The LIGF/LIG electrode was patterned on the PI film using infrared CO_2_ pulsed laser equipment from Universal Laser Systems [laser cutter platform VLS2.30 with high-power-density focusing optics (HPDFO 2.0, depth of focus is ± 0.76 mm)] in raster mode. The CO_2_ laser had a wavelength of 10.6 μm. The laser beam spot size was ~25 μm. The distance between the lens and the substrate was 54 mm. To obtain an LIGF/LIG electrode with the lowest sheet resistance, electrodes were fabricated under different laser powers ranging from 8.5 to 9 W in 0.1 W steps. The fixed parameters of laser scan speed, pulses per inch (PPI), and dots per inch (DPI) were 9 cm/s, 1000 PPI, and 500 DPI, respectively. All the CO_2_ laser irradiation was performed under ambient conditions.

### 2.3. LIGF/LIG Electrode-Based Glucose Sensor Fabrication and Cu NP Electroplating

The designed LIGF/LIG electrode-based glucose sensor (size = 14 mm × 22 mm) consists of a WE (diameter = 5 mm), a CE (area = 17.86 mm^2^), an RE (area = 2.1 mm^2^), an electrical path (1.5 mm × 10 mm), and pads (4 mm × 5 mm). The LIGF/LIG electrode-based glucose sensor fabricated under an 8.9 W laser power was used to investigate electrochemical behavior because it had the lowest sheet resistance, 6.8 Ω/□. The prepared LIGF/LIG electrode was cleaned sequentially with acetone, methanol, and distilled water. Next, the grown LIGF of the electrical paths and pads was removed with a strong N_2_ gas stream because the electrolyte solution could easily be flown into the electrical path and pad due to capillary force. A Kapton (PI) tape was attached to the electrical path area to define the electrochemical reaction area. Ag/AgCl ink was formed on the RE surface. The pads were covered with conducting silver epoxy to prevent damage to the pads due to physical force from the duck clip.

Copper NPs were electroplated on the WE by using the amperometric method [46]. This electrodeposition method has been reported to allow for the rapid and uniform deposition of copper nanoparticles in one step. Electroplating was conducted using a three-electrode system consisting of the LIGF/LIG electrode (WE), a graphite rod (CE), and a commercial Ag/AgCl electrode (RE). Electroplating was performed at −6 V for 10 sec in 80 mL of the electroplating solution (0.3 M CuSO_4_). To allow the Cu NPs to oxidize into the Cu_x_O form, cyclic voltammetry (CV) was performed in a 0.1 M NaOH solution at the applied range of −0.5 to +0.5 V at a scan rate of 50 mV/sec for 40 cycles. Figure 1 shows a schematic of the sensor fabrication process.

### 2.4. Electrochemical Characterization LIGF/LIG Electrode-Based Glucose Sensor in Original and Bent States

CV with a 20 mV/sec scan rate and 0–0.7 V scan range was performed at glucose concentrations of 0, 1, 2, 3, and 4 mM in 0.1 M NaOH aqueous solution. Chronoamperometry was recorded at +600 mV at different glucose concentrations, where 250 µL of glucose solution was added to the 0.1 M NaOH aqueous solution using a pipette. The selectivity of the fabricated LIGF/LIG-electrode-based glucose sensor was also evaluated by chronoamperometry under the same conditions by first adding 0.3 mM glucose solution and then adding 0.3 mM AA, UA, lactose, fructose, sucrose, and glucose solutions in sequence. Furthermore, the fabricated LIGF/LIG-electrode-based glucose sensor was also tested by the same electrochemical measurement methods (CV and chronoamperometry) after being bent at angles of 0°, 45°, 90°, 135°, and 180°.

## 3. Results and Discussion

### 3.1. Material Characterization

To optimize the fabrication of the LIGF/LIG electrode, we experimentally investigated the sheet resistance (*R*_s_) of electrodes fabricated using different laser powers (8.5 W–9 W with 0.1 W increment) at a 9 cm/s scan rate, 1000 PPI, and 500 DPI. As shown in Figure 2A, when the laser power applied to the PI film increased from 8.5 to 8.9 W, the measured *R*_s_ value of the LIGF/LIG electrode decreased. The LIGF/LIG electrode and PI film fabricated at a 9.0 W laser power were damaged by excessive thermal power, resulting in a physically unstable LIGF/LIG electrode and high *R*_s_. When an 8.9 W laser power was used, the lowest LIGF/LIG electrode sheet resistance of 6.8 Ω/□ was obtained due to its high carbon composition (98.5% atomic percentage of carbon, 1.3% atomic percentage of oxygen, 0.25% atomic percentage of nitrogen), which is in contrast with the electrodes fabricated at different laser powers (Figure 2B, Appendix A).

The 8.9 W LIGF/LIG electrode was also analyzed using XRD analysis and Raman spectroscopy. In the XRD pattern (Figure 2C), a dominant peak is observed at 26.4°, representing the (002) hexagonal graphitic crystal plane. The full width at half-maximum was relatively broad (7.81°), indicating the unusual ultra polycrystalline nature due to thermal expansion caused by laser irradiation [29]. The value of the interplanar distance (*d*) in the lattice planes calculated by Bragg’s law (2*d*·sin*θ* = *nλ*) is 3.37 Å, which is in good agreement with that of the graphitic phase. A small peak of the (100) graphitic crystal phase is also observed at 44.4°. Figure 2D shows the Raman spectrum of the LIGF/LIG electrode, which confirms the presence of the D, G, and 2D peaks. The Raman spectrum is similar to that of porous LIG, which shows the D, G, and 2D peaks at 1348.2, 1576.5, and 2694.0 cm^−1^, respectively. The observed D peak is caused by structural edge defects, bent sp^2^ carbon, or symmetry-broken sites in the LIGF and LIG, and the sharp 2D peak represents 2D graphite containing randomly stacked graphene layers [29,30].

To investigate their morphology, the Cu NP-electroplated 8.9 W LIGF/LIG electrodes were observed by FE-SEM. Figure 3A, B show the top view of the Cu NP-coated LIGF/LIG electrode that LIGF grew at 8.9 W over the porous LIG layer. Figure 3C depicts the Cu NP-coated LIGF at higher magnification (×30 k). Highly dense Cu NPs were uniformly electroplated on the LIGF and LIG surfaces. Furthermore, the presence of the electroplated Cu NPs was confirmed by energy-dispersive X-ray spectroscopy (EDS), as shown in Figure 3D. The weight % of Cu (copper), O (oxygen), and C (carbon) were measured to be 85.64, 9.45, and 4.91, respectively. A strong Cu peak appeared dominantly in the spectra of the Cu NP-coated LIGF/LIG electrode, indicating that the Cu NPs were successfully electroplated on the LIGF/LIG electrode (Appendix A). In addition, the atomic % of Cu shows almost twice the higher value than the atomic % of O. It might imply that the presence of metallic copper and Cu (I) oxide and Cu (II) oxide on the modified electrode during electroplating. Figure 3E,F show the cross-sectional view of a Cu NP-coated LIGF/LIG electrode. The measured thickness is ~420 μm. In addition, we also inspected bare LIGF and porous LIG at higher magnifications. LIGF with diameters ranging from 140 to 282 nm were grown. The pore diameters in the porous LIG were approximately 700 nm to 1 µm (Appendix A).

### 3.2. Electrochemical Characterization of Bare LIGF and Cu NP/LIGF Electrodes by CV

CV experiments using bare LIG, Cu NP/LIG, bare LIGF/LIG, and Cu NP/LIGF/LIG electrodes were conducted in [10 mM K_3_[Fe(CN)_6_] + 1 M KNO_3_] (1:1) ferricyanide solution at different scan rates from −0.4 to + 0.8 V to investigate the electrochemical reaction (Appendix A). The redox reaction was observed in both CV experiments. The peak current increased linearly in proportion to the square root of the scan rate, indicating a quasi-reversible surface electrode process. To determine the electrochemically active surface area and diffusion coefficient of the bare LIG, Cu NP/LIG, bare LIGF/LIG, and Cu NP/LIGF/LIG electrodes, the Randles–Sevcik plots were obtained, as shown in Figure 4. Simply put, the slope of the LIGF/LIG electrode is higher than that of the bare LIG electrode, which implies that the former has a higher surface area. These electrochemical parameters at RT can be determined using the Randles–Sevcik equation [47,48]:*I_p_* = (2.69 × 10^5^) *A* × *D*_0_^1/2^*n*^3/2^*v*^1/2^*C*(1)
where *A* is the area of the working electrode, *D*_0_ is the standard diffusion coefficient of K_3_Fe(CN)_6_ (i.e., 7.6 × 10^−6^ cm^2^/s), *n* is the number of electrons transferred in a redox event, *v* is the scan rate, and *C* is the concentration of the redox solution. In addition, a modified Randles–Sevcik equation can be expressed as the slope of the linear fitting line:*Slope* = (2.69 × 10^5^) *A × D*^1/2^*n*^3/2^*v*^1/2^*C*(2)
where *D* is the diffusion coefficient of the analyte. Therefore, using the constant values of the parameters *D*, *C*, and *n*, we could successfully obtain approximate values of *A* and *D* according to the Randles–Sevcik equation.

According to these numerical analyses in Table 1, the LIGF/LIG electrode has an active surface area of 0.372 cm^2^. This calculated result is larger than 0.197 cm^2^ of the bare LIG electrode and indicates that the grown LIGF can increase the active surface area. In addition, the active surface area of Cu NP/LIGF/LIG is 0.596 cm^2^. This result is 1.6 times larger compared to the LIGF/LIG electrode, which indicates that the Cu NPs are effectively electroplated on the LIGF and LIG. The diffusion coefficient values of the bare LIGF/LIG and Cu NP/LIGF/LIG electrodes are also calculated to be 228.35 cm^2^/s and 228.84 cm^2^/s, respectively. The *D* values are almost the same owing to the lack of reaction with the [10 mM K_3_[Fe(CN)_6_] + 1 M KNO_3_] (1:1) ferricyanide solution.

### 3.3. Electrochemical Characterization of Cu NP/LIGF/LIG Electrode for Glucose Detection

The CV curves of the fabricated Cu NP/LIGF/LIG electrode-based glucose sensor were obtained under 100 mV/s at a scan rate from 0 to +1 V using a 0.1 M NaOH solution with different glucose concentrations (Figure 5A). The CV oxidation peak current at +600 mV was found to increase in proportion to the glucose concentration, resulting in a high linear correlation constant (i.e., *R*^2^ = 0.997, *y_I_*_peak_ = 125.2*x* + 504, where *x* is the glucose concentration), as shown in the inset of Figure 5A. This good linearity of the measured current value demonstrates the electrochemical feasibility of the Cu NP/LIGF/LIG electrode as a glucose sensor. The mechanism for the oxidation of glucose by the Cu NP/LIGF/LIG electrode in the 0.1 M NaOH solution is as follows [49,50]:Cu + 2OH^−^ → CuO + H_2_O + 2e^−^(3)
orCu + 2OH^−^ = Cu(OH)_2_ + H_2_O + 2e^−^(4)

Then, CuO is further oxidized into Cu(III) species such as CuOOH^−^ or Cu(OH)_4_^−^ [51]:CuO + OH^−^ = CuOOH + e^−^(5)
orCuO + H_2_O + 2OH^−^ = Cu(OH)_4_^−^ + e^−^(6)

Next, glucose is catalytically oxidized by the Cu(III) species, forming hydrolyzate gluconic acid, and the Cu(III) compounds return to Cu(II) compounds and restart the process described in Equations (5) and (6) [52]:Cu(III) + R_1_–CHOH–R_2_ = R_1_–CHO–R_2_ + Cu(II)(7)

The amperometric current response was obtained by a chronoamperometry experiment at +600 mV in which 250 μL of glucose at different concentrations [50 μM (twice), 100 μM (nine times), and 1 mM (three times)] were pipetted into 10 mL of 0.1 M NaOH solution, and the amperometric current response was continuously measured for the 1-, 2-, 3-, and 4 mM glucose solutions (Figure 5B). With increasing glucose concentration, the response current increased and exhibited an excellent linear relationship in concentration ranges of 0.124–9.653 μM and 1–4 mM. This linear relationship can be expressed as *y* (μA) = 28.02*x* + 3.412 (*R*^2^ = 0.998) and *y* (μA) = 0.126*x* + 625 (*R*^2^ = 0.997), respectively (Figure 5C). The limit of detection (LOD) was estimated to be approximately 124 nM (S/N = 3.3), and the corresponding detection sensitivity was calculated as 1438.8 µA/mM∙cm^2^. The selectivity of our fabricated Cu NP/LIGF/LIG-electrode-based glucose sensor was also tested. The anti-interference property is crucial for the development of a non-enzymatic electrochemical biosensor because species such as AA, DA (dopamine), UA, and other easily oxidized compounds always coexist with glucose in human blood. A chronoamperometry experiment was performed using 250 μL of 0.3 mM ascorbic acid (AA), uric acid (UA), lactose, fructose, and sucrose in 10 mL of 0.1 M NaOH solution to represent human serum (Figure 5D). The results show no significant current signal other than the glucose detection signal, indicating that our Cu NP/LIGF/LIG-electrode-based glucose sensor has excellent selectivity for glucose sensing.

### 3.4. Electrochemical Characterization of Cu NP/LIGF/LIG-Electrode-Based Glucose Sensor Under Bending at 0°, 45°, 90°, 135°, and 180°

We also investigated the sheet resistance and CV characteristics of the Cu NP/LIGF/LIG electrode-based glucose sensor under bending at different angles using a rod with a curvature radius of 1 cm (Figure 6). Figure 6A–E show the bending conditions (0°, 45°, 90°, 135°, and 180°) under which the electrochemical properties were investigated. As shown in Figure 6F, the sheet resistance increases linearly with increasing bending angle because the porous LIG network would be stretched, resulting in increased *L* (which is calculated as *R* = *ρ·L*/*A*). In addition, physical defects resulting from bending of the electrode could be attributed to the increased sheet resistance. The sheet resistance of these bent Cu NP/LIGF/LIG electrodes can also affect their electrochemical properties. Consequently, it led to a reduction in the oxidation peak current signal (Figure 6G) by <5% (45°), 10% (90°), 13% (135°), and 18% (180°). (The raw data are given in the Appendix A). These results show good electrical properties owing to the current response signals of several hundred microamperes despite the increased sheet resistance; in addition, our fabricated Cu NP/LIGF/LIG electrode-based glucose sensor shows potential for use as a highly flexible glucose sensor due to its electrical stability.

To further characterize the amperometric current response at different bending angles, our fabricated Cu NP/LIGF/LIG electrode-based glucose sensor was tested with amperometry at +600 mV at 5 different angles: 0°, 45°, 90°, 135°, and 180°. Figure 7A shows the amperometric current results along with the time of applying different glucose concentrations for these 5 different bending angles. The different glucose concentrations have been produced by pipetting 250 μL of different concentrations of glucose [50 μM (twice), 100 μM (nine times), and 1 mM (three times)] into 10 mL of 0.1 M NaOH solution for the results of Figure 7A. With increasing bending angle, the current from the fabricated sensor generally decreased due to the increased sheet resistance. Nevertheless, these current responses show excellent linearity and a high current signal in a concentration range of 0.124–9.653 μM, as shown in Figure 7B. The sensitivity under each bending condition is summarized in Table 2. Many non-enzymatic glucose sensors based on Cu nanostructures have been reported. The performance of our fabricated Cu NP/LIGF/LIG electrode-based glucose sensor, including the sensitivity, LOD, and linear range, is compared in Table 2 with those of other cupric oxide and carbon material-based non-enzymatic glucose sensors. From comparative data, it can be deduced that the Cu/LIGF/LIG electrode exhibits notable performance indicators such as high sensitivity, a low detection limit, and good linearity, even in the bent state. In particular, the Cu/LIGF/LIG electrode can provide a highly electrochemically active surface area due to its fibers and porous network, which provide many electroactive sites and improve the electron transfer rate owing to direct electron delivery to the electrode.

### 3.5. Stability of the Fabricated Cu NP/LIGF/LIG Electrode-Based Glucose Sensor Under Bending Angles of 0°, 45°, 90°, 135°, and 180°

The long-term stability of the fabricated Cu NP/LIGF/LIG electrode under bending angles of 0°, 45°, 90°, 135°, and 180° was tested by measuring the current response in the presence of 1 mM glucose every two days for 15 days (Figure 8). The fabricated Cu NP/LIGF/LIG electrode could maintain within 10% of its original sensitivity, indicating its excellent chemical stability.

## 4. Conclusions

In summary, an LIGF/LIG electrode was directly fabricated on PI film by a simple and fast method of laser patterning in raster mode. The sheet resistance and thickness of the fabricated LIGF/LIG are 6.8 Ω/□ and ~420 μm, respectively. Furthermore, basic electrochemical analysis using Randles–Sevcik plots of the bare LIGF/LIG and Cu/LIGF/LIG electrodes shows good linearity and a large active surface area. In addition, a Cu NP/LIGF/LIG electrode-based glucose sensor fabricated by a conventional Cu electroplating method shows excellent glucose detection characteristics such as a high sensitivity of 1438.8 µA/mM∙cm^2^, a low detection limit of 124 nM, and a broad linear range at an applied potential of +600 mV. Significantly, the resulting Cu NP/LIGF/LIG electrode-based glucose sensor bent at 0°, 45°, 90°, 135°, and 180° achieved relatively high sensitivity, low LOD, and good linear detection range.

These results indicate that our work offers an inexpensive, facile method to fabricate non-enzymatic glucose sensors and have demonstrated their electrochemical properties under different bending conditions. In addition, our fabricated flexible Cu NP/LIGF/LIG electrode-based glucose sensor can be applied to various bioelectronic devices such as wearable devices and skin electronics. In the near future, we will demonstrate the possibility of using our LIGF/LIG electrode-based multisensor for detecting glucose, lactate, K^+^, and Na^+^ contents and temperature of human sera such as saliva, sweat, or tears.

## Figures and Tables

**Figure 1 sensors-25-02341-f001:**
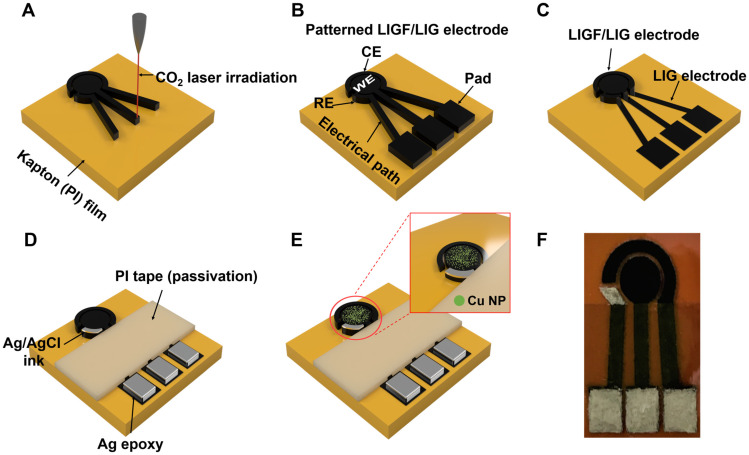
Schematic of the fabrication process: (**A**) CO_2_ irradiation of PI film to form the LIGF/LIG electrode pattern, (**B**) fabricated LIGF/LIG electrode-based sensor, (**C**) removal of LIGF from the electrical path and pad with N_2_ gas stream, (**D**) formation of Ag/AgCl electrode in RE area and preparation of the passivation layer with PI tape and Ag epoxy, (**E**) completed Cu NP/LIGF/LIG electrode-based glucose sensor obtained by electroplating Cu NPs on the WE area, (**F**) photograph of the fabricated Cu NP/LIGF/LIG electrode-based glucose sensor.

**Figure 2 sensors-25-02341-f002:**
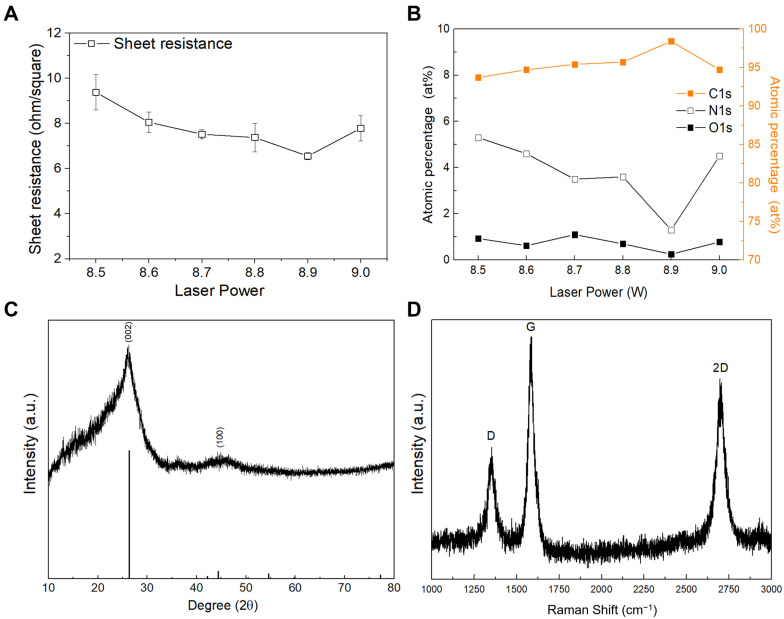
(**A**) Measured sheet resistance and (**B**) XPS results showing atomic percentages of C1s, O1s, and N1s of the fabricated LIGF/LIG electrodes at laser powers of 8.5, 8.6, 8.7, 8.8, 8.9, and 9 W. (**C**) XRD pattern and (**D**) Raman spectrum of the fabricated LIGF/LIG electrode at 8.9 W.

**Figure 3 sensors-25-02341-f003:**
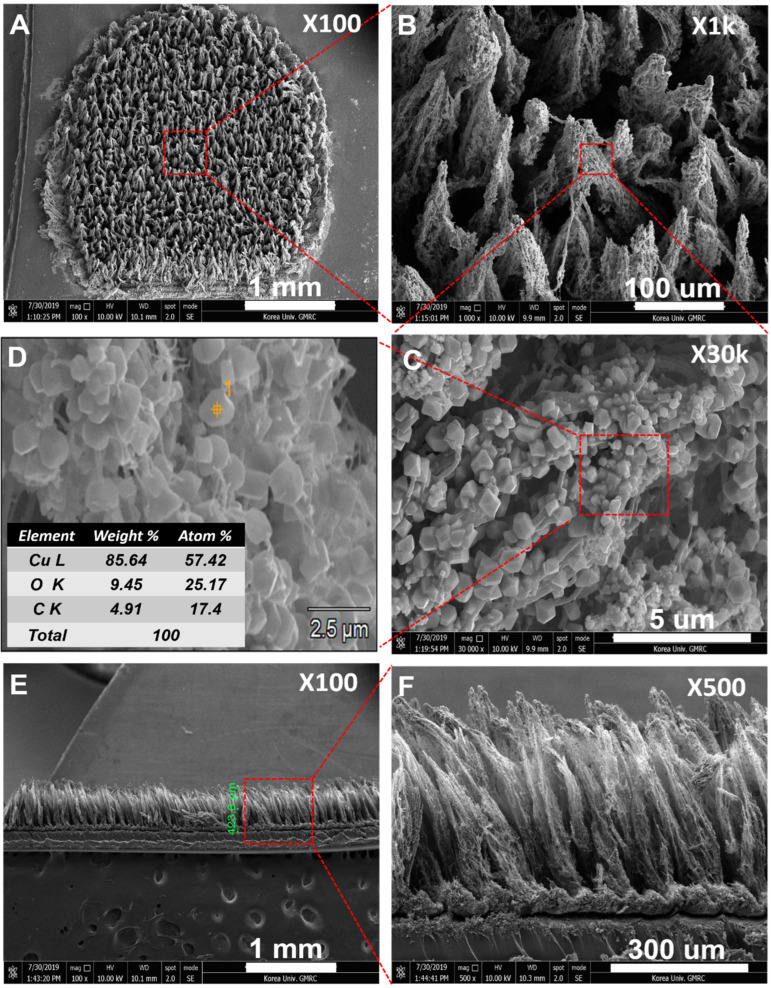
FE-SEM images of the top view of (**A**) the fabricated Cu NP/LIGF/LIG electrode (×100), (**B**) Cu NP-coated LIGF (×1k), (**C**) Cu NP-coated LIGF (×30k) at higher magnification, (**D**) EDS results for the Cu-NP-coated LIGF/LIG electrode, and (**E**,**F**) a cross-sectional view of the LIGF/LIG electrode fabricated under an 8.9 W laser power.

**Figure 4 sensors-25-02341-f004:**
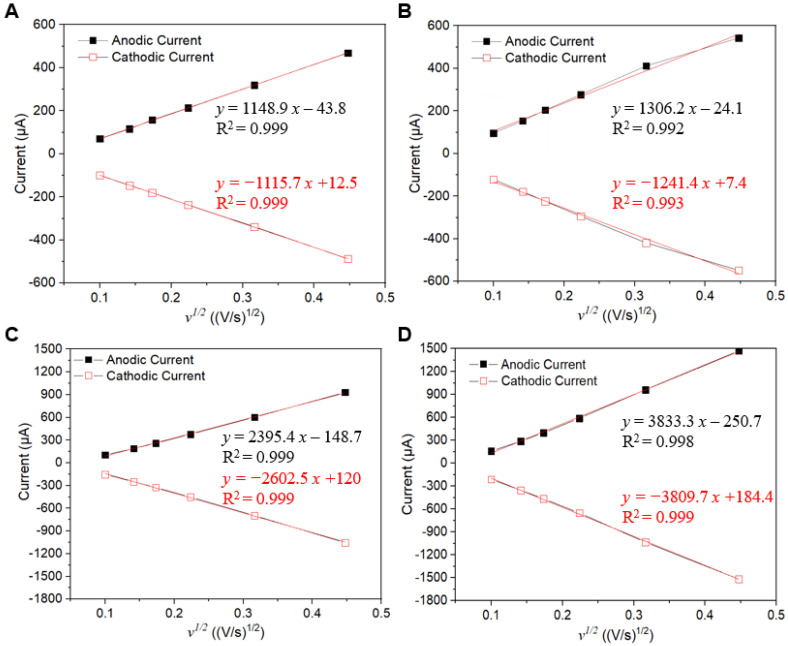
Electrochemical characterization of the bare LIG, Cu NP/LIG, the bare LIGF/LIG, and Cu NP/LIGF/LIG electrodes in [10 mM K_3_[Fe(CN)_6_] + 1 M KNO_3_] (1:1) ferricyanide solution. Scan rate: 10, 20, 30, 50, 100, and 200 mV/s. Randles–Sevcik plots with linear fitting curves of (**A**) bare LIG electrode, (**B**) Cu NP/LIG electrode, (**C**) bare LIGF/LIG electrode, and (**D**) Cu NP/LIGF/LIG electrode.

**Figure 5 sensors-25-02341-f005:**
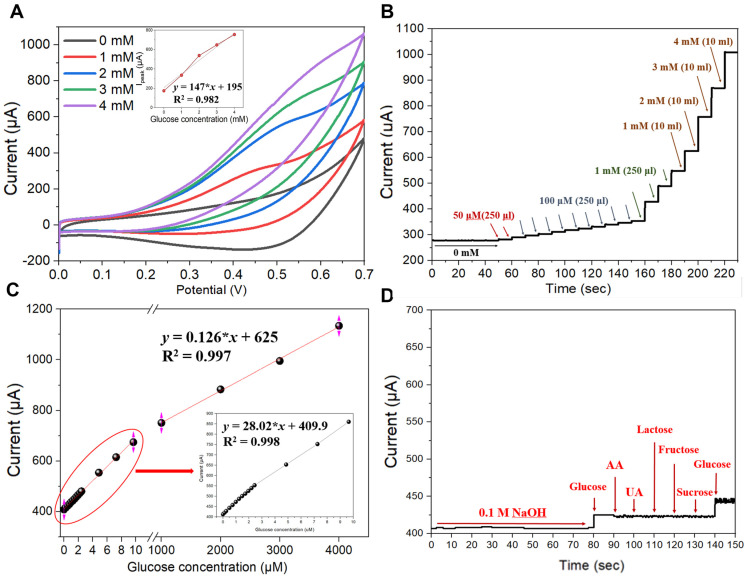
(**A**) CV curves of the Cu NP/LIGF/LIG electrode at glucose concentrations of 0, 1, 2, 3, and 4 mM in 10 mL of 0.1 M NaOH solution. (**B**) Amperometric current response at +600 mV; glucose concentration was increased every 10 s by pipetting 250 μL of glucose at different concentrations into 10 mL of 0.1 M NaOH solution [50 μM (twice), 100 μM (nine times), and 1 mM (three times) in sequence], and then the amperometric current response was measured for 1, 2, 3, and 4 mM of glucose in 10 mL of 0.1 M NaOH solution. (**C**) Linear fitting curve from amperometric current response results. (**D**) Selectivity test using chronoamperometry, where 250 μL of 0.3 mM solutions of glucose, AA, UA, lactose, fructose, sucrose, and glucose were pipetted sequentially into 10 mL of 0.1 M NaOH solution.

**Figure 6 sensors-25-02341-f006:**
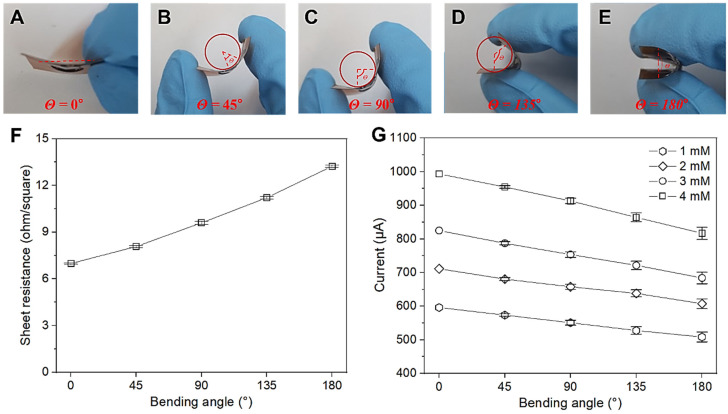
Cu NP/LIGF/LIG electrode-based glucose sensor in bent states with different bending angles (*θ*) for sheet resistance and CV measurements: (**A**) 0°, (**B**) 45°, (**C**) 90°, (**D**) 135°, and (**E**) 180°. In (**B**–**E**), the curvature radius is 1 cm. (**F**) Sheet resistance and (**G**) CV results of Cu NP/LIGF/LIG-electrode-based glucose sensor at glucose concentrations of 1, 2, 3, and 4 mM for each bending angle.

**Figure 7 sensors-25-02341-f007:**
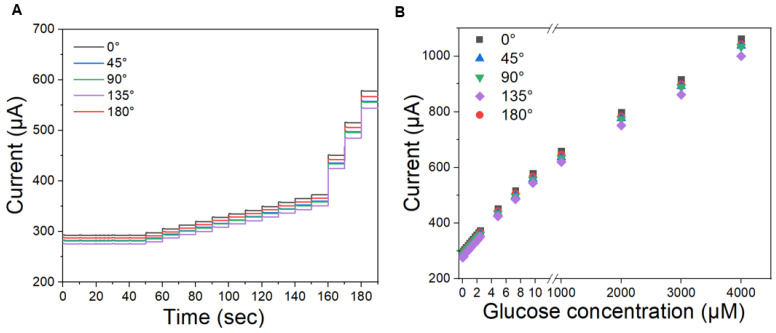
(**A**) Amperometric current response at +600 mV with time of applying different glucose concentrations for five different bending angles and (**B**) amperometric current response with respect to glucose concentration to investigate its linearity at bending angles of 0°, 45°, 90°, 135°, and 180°.

**Figure 8 sensors-25-02341-f008:**
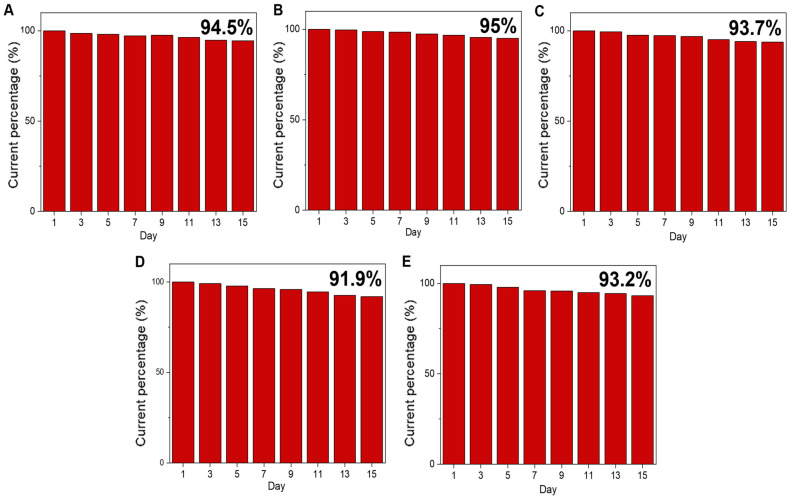
Stability of the fabricated Cu NP/LIGF/LIG electrode stored at ambient conditions over 15 days with the addition of 1 mM glucose under bending state at (**A**) 0°, (**B**) 45°, (**C**) 90°, (**D**) 135°, and (**E**) 180°.

**Table 1 sensors-25-02341-t001:** Comparison of active surface area, diffusion coefficient, and correlation coefficient (R^2^) of the bare LIG, Cu NP/LIG, bare LIGF/LIG, and Cu NP/LIGF/LIG electrodes.

		bare LIG	Cu NP/LIG	bare LIGF/LIG	Cu NP/LIGF/LIG
Active surface area (cm^2^)		0.197	0.254	0.372	0.596
Diffusion coefficient (cm^2^/s)		112.56	145.49	228.35	228.84
R^2^	Anodic current	0.999	0.992	0.999	0.998
	Cathodic current	0.999	0.993	0.999	0.999

**Table 2 sensors-25-02341-t002:** Comparison of the performance of Cu-nanostructure-based glucose sensors.

Modified Electrode		Sensitivity (µA/mM∙cm^2^)	LOD (µM)	Linear Range (mM)	Ref.
Cu NP/GE		-	0.4	0.001–0.2	[53]
Cu NF/PIGE		442.1	0.39	1–19	[54]
Cu NR/GE		371.4	4	-	[55]
Cu NF/GE		709.5	4	0.004–8	
Cu NF/MWCNT-Graphite		1211	4	0.004–14.5	[56]
Cu NP/PGE		1467.5	0.44	-	[57]
CuO/LEG		619	0.049	0–3	[58]
Cu/LIG		281.7	6.1	0.5–4	[59]
Cu NP/LIG/paper		87.6	14	0.02–1.51	[60]
Cu NP/LIGF/LIG	0°	1438.8	0.124	0–4	This work
	45°	1415.7			
	90°	1402.2			
	135°	1390.5			
	180°	1422.1			

NP: nanoparticle, NF: nanofiber, NR: nanorod, GE: graphite electrode, PIGE: paraffin-impregnated graphite electrode, PGE: pencil graphite electrode, LEG: laser-engraved graphene.

## Data Availability

Data are contained within the article.

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
