# Peer review of "Fabrication and Characterization of a Flexible Non-Enzymatic Electrochemical Glucose Sensor Using a Cu Nanoparticle/Laser-Induced Graphene Fiber/Porous Laser-Induced Graphene Network Electrode"

_sensors, 2025, doi:10.3390/s25072341_

Round 1

Reviewer 1 Report

Comments and Suggestions for Authors

Taeheon Kim and James Jungho Pak reported a Cu/LIGF/LIG electrode-based glucose sensor. The sensor shows excellent glucose detection characteristics and great flexibility, which will promote the development of flexible sensors. The manuscript is well organized and well written. Thus, I recommend it to be published after addressing these issues:

  1. Why do you emphasize “non-enzymatic” glucose sensor? This should be explained in Introduction.
  2. To my knowledge, Cu nanoparticles are very easy to be oxidized. Thus, the final components may not be Cu NPs but copper oxide NPs. According to element percentage (Figure 3D), the components seem to be Cu2O. If possible, further materials characterizations are suggested to performed to confirm the components, such as XRD, XPS, or TEM. According to sensing mechanism (equations 3-7), the direct sensing materials are copper oxides not copper.
  3. Many LIG-based electrochemical sensors are reported but not reviewed in Introduction, such as Non-modified laser-induced graphene sensors for lead ion detection, ACS Applied Nano Materials, 2023, 6(5), 3599-3607; Laser-scribed graphene sensors on nail polish with tunable composition for electrochemical detection of nitrite and glucose." Sensors and Actuators B: Chemical357 (2022): 131394. Also, electroplated LIG-based electrochemical sensors have been reported, such as A High-Performance Phosphate Potentiometric Sensor Based on Cobalt Nanoparticles Electroplated on Hierarchically Porous Laser-Induced Graphene. Advanced Materials Interfaces. 2024, 2301111.

Reviewer 2 Report

Comments and Suggestions for Authors

The manuscript is devoted to the  development of  a flexible electrochemical biosensor for non-enzymatic glucose detection based on screen printed electrode modified with Cu nanoparticle /graphene fiber (LIGF)/porous graphene (LIG) network structure on a polyimide film. The idea to use Cu NP in nonezymatic glucose sensor is not new, but authors combine it with LIGF and LIG network and obtained flexible sensor with good selectivity and sensitivity to glucose. 

The material is of practical and theoretical interest. The manuscript can be recommended for publication in the journal Sensors after  revision.

Items needed to be clarified:

Where authors plan to detect trace amounts of glucose ? Usually it is macro component in biological objects and food. It is more important to detect high amounts of glucose in blood from 1 mM up to 10 mM with high precision. 

Reviewer 3 Report

Comments and Suggestions for Authors

Comments: This article introduces the manufacturing and characterization of a flexible non enzymatic electrochemical glucose sensor, which detects glucose under different bending conditions. Overall, this job has certain practical application value. But there are still some small questions that need to be answered.

1 The quality of the pictures needs improvement, as some of the handwriting in the pictures is too small.

2 Comparison with other sensors: The document mentions that the performance indicators of Cu/LIGF/LIG electrodes are superior to other sensors, but does not provide detailed comparative data or charts.

3 Cost and preparation process: Although it is mentioned that Cu nanoparticles have the advantage of low cost, the preparation cost and process complexity of the entire sensor are not explained in detail.

4 Some recent related work should be discussed to draw more readers’ attention. Such as 10.1007/s12598-023-02318-9; 10.1016/j.jmat.2020.02.002; 10.1016/j.esci.2023.100133; 10.20517/ss.2023.17
